# Gene Duplication and Gene Fusion Are Important Drivers of Tumourigenesis during Cancer Evolution

**DOI:** 10.3390/genes12091376

**Published:** 2021-08-31

**Authors:** Cian Glenfield, Hideki Innan

**Affiliations:** Department of Evolutionary Studies of Biosystems, SOKENDAI, The Graduate University for Advanced Studies, Shonan Village, Hayama, Kanagawar 240-0193, Japan; glenfiec@tcd.ie

**Keywords:** tumour evolution, gene duplication, gene amplification, gene fusion, cancer genome, genome rearrangement

## Abstract

Chromosomal rearrangement and genome instability are common features of cancer cells in human. Consequently, gene duplication and gene fusion events are frequently observed in human malignancies and many of the products of these events are pathogenic, representing significant drivers of tumourigenesis and cancer evolution. In certain subsets of cancers duplicated and fused genes appear to be essential for initiation of tumour formation, and some even have the capability of transforming normal cells, highlighting the importance of understanding the events that result in their formation. The mechanisms that drive gene duplication and fusion are unregulated in cancer and they facilitate rapid evolution by selective forces akin to Darwinian survival of the fittest on a cellular level. In this review, we examine current knowledge of the landscape and prevalence of gene duplication and gene fusion in human cancers.

## 1. Introduction

Gene duplication is thought to be one of the predominant processes by which new genes and genetic novelty arise throughout evolution [1,2,3,4,5], and much of the same hallmark structural variation that facilitates genomic evolution over long periods of time, including gene duplication and gene fusion, can also be observed occurring rapidly in cancer during tumour formation. Cancer is a disease characterised by rapid proliferation and spread of clonal somatic cells within a tissue. To become tumourigenic, clones must evolve the ability to ignore regulatory pathways that normally place strict constraints on cell division and growth, increase the rate at which its cells proliferate and survive within a given micro-environment, and escape immune system surveillance [6].

Despite our progressively increasing knowledge of the mechanisms behind cancer formation and evolution the collective diseases remain the second leading cause of death worldwide, with nearly 10 million fatalities as a result of cancer in 2020 [7]. Genome abnormalities, including chromosomal rearrangement, changes in copy number, and single nucleotide mutations, are commonly observed in cancers, with many of these abnormalities serving as drivers of tumourigenesis [8]. Cancer cells undergo dynamic clonal expansion and experience an increase in genetic diversity in a given tissue micro-environment over a typically very short period of time [9]. Tumourigenesis and cancer progression could, therefore, be viewed as a microcosm of accelerated evolution. Indeed, the idea that cancer formation and progression can be considered an evolutionary process was first theorised in 1976 [10], a theory which has since been validated using modern genomics approaches.

Tumourigenesis has traditionally been thought to occur in a multistage step-wise fashion by gradual accumulation of genetic changes, accompanied by an increasing amount of chromosomal rearrangement and genomic instability in cancer cells [6,11]. Such instability can facilitate evolution by gene duplication, often termed gene amplification in cancer biology, which can be either small-scale, where a single gene or small group of genes are duplicated, or on a much larger scale, where the gene content of an entire chromosome or even the whole genome is doubled [8,12,13]. Additionally, frequent chromosomal rearrangement, tandem duplication, and deletion events in cancer cells can result in the joining together of two distinct genes to form a fusion gene, the products of which play an important role in tumour evolution and progression in many cancers [14,15].

In this review, we discuss and examine how gene duplication and gene fusion contribute to tumourigenesis and acquisition of therapeutic resistance in human cancers.

## 2. Routes to Gene Duplication in Cancer

Tumour formation and evolution can occur as a result of different mechanisms—including point mutations and structural alterations, including oncogene duplication—and common belief is that multiple aberrations are usually required to initiate tumourigenesis [16,17,18]. It is well documented that structural variation is capable of driving tumour formation and can result in the duplication and amplification of genes that may enhance cancer cell proliferation and survival relative to wild-type normal cells, thereby enhancing the ability of these cancerous cells to expand and further colonise their intratissue and extratissue environments [6,9]. Such gene duplication is very common in many human cancers and contributes to tumourigenesis usually due to the overexpression of oncogenes [19,20]. Additionally, cancers can acquire chemotherapeutic resistance to certain anti-cancer drugs as a result of overexpression of certain genes via duplication [21]. Indeed, the first cancer gene amplification to be identified, in 1978, was the dihydrofolate reductase (*DHFR*) gene; DHFR is inhibited by the anti-cancer drug methotrexate, leading to reduced cell growth and proliferation in vulnerable cells. However, duplication of *DHFR* provides a selective advantage by greatly enhancing DHFR synthesis and facilitating expansion of methotrexate-resistant tumour clones [22].

There are a variety of mechanisms by which—in large part due to increased genomic instability and formation of double-strand breaks—oncogenes can become duplicated in human cancers, including intrachromasomal deletion and tandem duplication, breakage-fusion-bridge events, or the formation of neochromosomes in certain tumour types [11,23,24,25,26]. Recent evidence shows that aberrant long interspersed nuclear element (LINE-1) retrotransposon integration in cancer genomes can also mediate large-scale duplication of chromosomal regions [27]. Duplicated genes can usually be found either in tandem arrays along a chromosome, contained within small, circularised extrachromosomal DNA (ecDNA) fragments, or interspersed randomly across the genome in intrachromosomal homogenously staining regions (HSRs) [19,28]. Additionally, certain hotspots may exist in cancer genomes that give rise to a greater frequency of tandem duplication of oncogenes, including *MYC* and *CCND1* [29]. Cancer associated ecDNAs are generally large (>1 Mb) and contain one or more full genes and regulatory regions which are often oncogenic, and a subset of ecDNAs (~30%) form pairs of chromatin bodies, termed double minutes (DMs) [20,21,30]. Once DMs and ecDNA fragments arise they are capable of reintegrating into chromosomal DNA at or near telomeres, which further propagates genomic instability. Furthermore, unequal segregation of ecDNAs from parent to daughter tumour cells can be a powerful driver of tumour evolution by rapidly increasing intratumour heterogeneity, allowing some daughter cells to posses many more ecDNAs than their parents, which could confer additional selective advantages [31,32,33,34]. Although oncogenic DMs and ecDNAs are important players in cancer pathogenesis, ecDNA fragments are not unique to cancer cells; smaller ecDNA fragments (200–500 bp) lacking full genes can be found in normal somatic tissues across different eukaryotic species—including human, *Caenorhabditis elegans*, and *Drosophila melanogaster*—as well as in the germline genomes of a subset (~0.5%) of human individuals [35,36,37,38,39]. However, the function and evolutionary relevance of ecDNAs in healthy somatic and germline tissues remains poorly understood.

### 2.1. Chromothripsis

In cancer, duplication and amplification events are sometimes preceded and facilitated by chromothripsis, which is a massive, catastrophic breakage and random rearrangement of genomic DNA encompassing one or several chromosomes that occurs in a single event (Figure 1) [40,41,42]. Evidence suggests that in many cases chromothripsis occurs early in tumourigenesis and quickly provides a huge selective advantage, which lends credence to the idea that in many cases cancer evolution may proceed in a series of rapid events in a short period of time before stabilising somewhat, in a manner akin to punctuated equilibrium and in contrast to the commonly accepted theory of evolutionary gradualism [9,18,43,44,45,46,47]. These observations are contrary to the idea that chromothripsis occurs as a result of cancer genome instability, when in fact reverse causation appears to be true for many tumours. Indeed, in recent years the two-phased model of cancer evolution—involving a punctuated phase and a stepwise phase—is gaining more acceptance, whereby the punctuated phase involves large, macroevolutionary events, such as chromothripsis facilitating rapid advances in tumour evolution, followed by the stepwise phase where cells featuring advantageous macro-evolutionary events grow and expand clonally and experience smaller-scale gene-based mutation [13,48,49,50]. These large genome-rearrangement events change the karyotype coding of the cell—essentially the genomic organisation and system-level information—and it is thought that this can substantially influence subsequent cancer evolution by altering gene expression and interaction networks [51].

The role and frequency of chromothripsis in cancer is becoming more apparent, but the mechanisms responsible for its initiation remain poorly understood. One of the most attractive models explaining chromothripsis is the micronuclei model, whereby mitotic errors in cells with defective chromosomal segregation during the transition from metaphase to anaphase can result in the formation of micronuclei, which can contain whole chromosomes or chromosome fragments [52,53]. After these micronuclei experience defective and asynchronous DNA replication the DNA damage response pathway is activated, but the DNA repair and cell cycle checkpoint pathways subsequently fail to activate and the incorrectly replicated micronuclei become fragmented. These fragments can be rearranged to form a new chromosome, which is then reintegrated into the primary nucleus of a subsequent daughter cell [54]. Regardless of how chromothripsis is initiated, studies have shown that it can thus result in extensive genome rearrangement, gene amplification, and deletion of chromosomal regions in cancer cells, and evidence points to chromothripsis as an important mechanism driving formation of ecDNAs/DMs and fusion genes during tumourigenesis and acquisition of chemotherapeutic drug tolerance [13,55].

Selective pressure on cancer cells from chemotherapeutic intervention can drive further evolution of cancer genomes, resulting in enhanced genomic instability, intra- and extrachromosomal gene amplifications, and chromothripsis. Shoshani et al. [13] demonstrated that when cancer cells were subjected to variable levels of selection pressure—in the form of the anticancer drug methotrexate—weaker selection pressure was associated with low-level gain in copy number, whereas applying stronger selection resulted in formation of circular DMs derived from chromothripsis. They further showed that cancer cells can undergo subsequent rounds of chromothripsis in response to increasing selection pressure, leading to further amplification of DMs—which require non-homologous end joining for proper formation—and allowing the cells to develop tolerance to the drug by duplication and amplification of *DHFR*. These observations highlight the remarkable ability of chromothripsis to drive tumourigenesis and adaptation to unfavourable environmental conditions.

### 2.2. Whole-Genome and Whole-Chromosome Duplication

In addition to chromothripsis and small-scale intrachromasomal duplication, gene amplification in cancer can occur as a result of ploidy changes including tetraploidy/whole-genome duplication (WGD) and aneuploidy in the form of whole-chromosome duplication (WCD). WGD and aneuploidy are classic hallmarks of cancer, are incredibly common across diverse cancer types, and are significantly associated with poor prognosis for patients [56,57]. Despite the frequency of WGD in human cancers, its impact on tumourigenesis and the underlying mechanisms that cause it to occur still remain somewhat elusive. The presence of WGD and WCD in tumour cell populations (WGD${}^{+}$/WCD${}^{+}$) is thought to result from flawed or defective cell division and checkpoint activation: a failure of cytokinesis—which is normally tightly controlled—to correctly segregate chromosomes in dividing cells [12,58]. A variety of causes have been proposed to explain cytokinesis failure (reviewed extensively in Lens et al. [59]): physical obstruction by a number of factors such as the presence of asbestos fibres or chromatin that can disrupt cleavage furrow ingression; delayed mitosis giving rise to mitotic slippage, such that anaphase is not initiated and tetraploidisation occurs; or possible rare mutations in—or aberrant expression of—regulators of cytokinesis. Regardless of how it occurs, these macroevolutionary WGD/WCD events provide tumours with an abundance of adaptive potential and tumour cell diversity since all genes on a chromosome or in the genome are duplicated. Oncogenes that facilitate enhanced cellular proliferation and survival can, therefore, be upregulated while duplicated tumour suppressor genes can be selectively lost and returned to a diploid state, or lost entirely.

Evidence suggests that WGD, like chromothripsis, tends to occur early in cancer evolution, causing cells to become oncogenic and driving additional chromosomal instability (CIN) and aneuploidy, which in turn further facilitates tumourigenesis and intratumour heterogeneity [59,60]. Large regions of the newly tetraploid chromosomes can be subsequently lost after WGD, such as regions of chromosome arm 4q in colorectal cancer tumours; loss of these regions is significantly associated with increased genomic instability and worse prognosis for patients [61]. Although the majority of WGD${}^{+}$ clones are sub-tetraploid, the median ploidy of WGD${}^{+}$ advanced tumours was found to be 3.3, relative to a ploidy of 2–2.1 for WGD${}^{-}$ tumours, suggesting many of the doubled chromosome regions are retained after WGD [56,57]. Evolutionary simulations showed that WGD can be selected for in cancer to increase tolerance of aneuploidy and to act as a buffer against the accumulating deleterious effects of somatic mutations, loss of heterozygosity, and chromosomal aberrations [62]. Additionally, certain chromosomes are more frequently subject to WCD during tumourigenesis than expected by chance, including chromosomes 7, 12, and 20 [63]. These observations may indicate that there is selection for retention of these chromosomes in many tumours, possibly due to the enhanced cellular proliferation and survival benefits conferred by their duplicated gene content. However, experimental systems have shown that too much CIN can be detrimental to tumours, which suggests that a balance exists between the tumourigenesis-promoting effects of aneuploidy and its accompanying instability and fitness cost, much like the effects of any genomic change or innovation that arise during species evolution [64]. Furthermore, WGD appears to open up tumour cells to genetic vulnerabilities due to additional requirements for survival, such as the p53 tumour–suppressor pathway that becomes active in response to WGD, as well as the physiological fitness costs associated with WGD [65,66]. These vulnerabilities could be exploited to enhance cancer therapies.

## 3. Frequency of Structural Variation Leading to Duplication in Cancer

The importance of structural variants to tumourigenesis and cancer evolution can be summarised with the following statistic: between 68% to 80% of human solid tumours are aneuploid, possessing an abnormal number of chromosomes [59,63]. That said, certain types of cancers are more susceptible to undergoing structural variation and genome rearrangement during tumour evolution than others. For instance, the frequency of chromothripsis varies widely across cancer types, with a recent pan-cancer analysis of whole genomes (PCAWG) study estimating that, pan-cancer, chromothripsis occurs at a rate of ~29% (considering high-confidence events only); however, several cancer types experience a chromothriptic rate of >40%, with liposarcomas and osteocarcinomas reaching 100% and 77% occurrence, respectively [18,55]. These levels of chromothripsis are much higher than previous estimates, though this is primarily attributed to the higher sensitivity of the algorithms used by PCAWG [40,56,67,68,69]. By contrast, some cancer types experience chromothripsis infrequently—at rates <5%—including pilocytic astrocytomas and chronic lympocytic leukaemia, suggesting certain tumour environments or cellular states can be more or less favourable for chromothripsis. Intriguingly, at least 11.1% of focal amplifications involving oncogenes—including *CDK4*, *MDM2*, and *MYC*—localise to regions that have undergone chromothripsis and are frequently upregulated as a result, highlighting the importance of chromothripsis to gene amplification.

In a parallel PCAWG study of structural variation within 2559 tumour samples from 38 distinct tumour types, nearly 95% of samples (2429/2559) had at least one detectable somatic copy number alteration (SCNA) that was not present in their respective germline control samples [8,18]. The most common class of simple SCNA that was identified was deletion, followed by tandem duplication, with unbalanced translocations being less likely. Significant variability in the frequency and distribution of these different SCNAs was observed both between and within different tumour types. Ovarian cancer in particular appears to have a high frequency of tandem duplication or deletion. Additionally, this study also identified a subset of liver cancer samples that exhibit SCNAs resulting in duplication of the *Telomerase reverse transcriptase* (*TERT*) gene and enhancing its expression; *TERT* is a well known oncogene frequently duplicated in cancer and is responsible for extending telomeric DNA, thereby facilitating immortalisation of cancer cells by overcoming the Hayflick limit on the number of cell divisions [70,71,72,73,74]. Interestingly, while duplication usually contributes to tumourigenesis by enhancing oncogene expression, tandem duplication can also result in deactivation of tumour suppressor genes by disrupting exon open reading frames, such as *PTEN* in ovarian and breast tumours.

Aside from obvious cases involving oncogenes, such as *TERT*, disentangling gene amplifications that drive cancer evolution from those that arise as a result of passenger SCNA events remains a challenging task for cancer biologists, particularly since SCNAs can encompass many genes [75]. This is further complicated by the frequent presence of complex, secondary chromosomal rearrangements which can mask the initial copy number breakpoints that give rise to genomic amplifcations [76,77,78]. Pan-cancer, small-scale intrachromosomal focal amplifications, have been identified in regions that are duplicated recurrently across diverse cancer types. Some of these duplicated regions do not contain known oncogenes, but are significantly enriched for epigenetic regulatory genes suggesting an underappreciated role for these genes in cancer formation [56]. The recurrent nature of these duplicated regions suggests that they experience positive selection, which generally has an edge over purifying selection during tumourigenesis [79].

WGD is emerging as one of the most common pan-cancer genomic aberrations in human and is unsurprisingly associated with increased levels of SCNA. Bielski et al. [57] sequenced the tumours of 9692 patients with advanced cancers and observed WGD at a prevalence of almost 30%; these WGD${}^{+}$ cancers were associated with increased morbidity and decreased overall survival regardless of age, *TP53* mutation, or cancer type. Using data from The Cancer Genome Atlas (TCGA), Quinton et al. [66] found a WGD${}^{+}$ prevalence of ~36% across 10,000 primary tumour samples comprising 32 different tumour types, consistent with the above mentioned and previous studies [56]. Among metastatic cancers the rate appears to be even higher, with 56% of metastatic solid tumours exhibiting WGD events, highlighting the contribution of WGD to cancer progression and poor prognosis for patients [80]. Furthermore, tumours with mutant *TP53*—a potent tumour-suppressor gene that is thought to guard against WGD${}^{+}$ and aneuploid cells continuing through successive cell cycles and proliferating—were significantly associated with WGD${}^{+}$ (1.8-fold increase in frequency), as well as chromothripsis (1.54-fold increase) [55,81]. Consistent with the hypothesis that *TP53*-disabling mutations facilitate WGD and not vice versa, studies have shown that functional mutations in *TP53* precede WGD in >90% of unambiguous cases [82]. However, it is evident that mutant *TP53* is not required for WGD or chromothripsis to arise, as ~21% and ~24% of wild-type *TP53* tumours also exhibit WGD and chromothripsis, respectively. Similarly to chromothripsis, the purported occurrence of WGD in cancer varies substantially by tumour type and even molecular and histological sub-types. Germ cell tumours appear to most frequently exhibit WGD events, at a rate of ~58%, compared to <5% of gastrointestinal neuroendocrine tumours [57]. These observations suggest that certain cellular environments are more favourable or tolerant of WGD than others, and may be under contrasting selective forces in different tissues. In support of this idea, recent studies have shown that the frequency of aneuploidy in different cell types can be influenced by the immune system, which can often recognise cancer cells in early tumourigenic stages due to the presence of aneuploidy and genome instability [83,84]. Immune tolerance of aneuploidy could, therefore, depend on the tumour micro-environment and the presence or absence of certain immune cells, which may determine whether aneuploidy promotes either immune system recognition or evasion of nascent cancers [85,86,87] Nevertheless, WGD clearly represents a significant macroevolutionary driver event in a swath of human cancers.

## 4. Gene Fusions Give Rise to Genetic Novelty

Gene fusion is a process that involves the merging of distinct, independent genes or fragments of genes via chromosomal inversion, tandem duplication, interstitial deletion, or translocation events (Figure 2) [14,15]. As mentioned previously, gene duplication, deletion, and chromosomal rearrangement are common in cancer genomes, and these events often precede presentation of observable malignant phenotypes [6,88,89]. The first translocation to be described in human cancer was in 1960, where a reciprocal translocation between chromosomes 9 and 22 in chronic myeloid leukaemia (CML) was described, resulting in the formation of the Philadelphia (Ph) chromosome (Figure 2A), an unusually shortened version of chromosome 22 [90,91]. This discovery subsequently led to the molecular elucidation of the first gene fusions and translocations in the early 1980s: fusion of *ABL1* and *BCR* resulting from the formation of the Ph chromosome causes constitutive activation of a tyrosine kinase in CML (Figure 3A); and a translocation in ~80% of Burkitt lymphomas that places the *MYC* gene within a region of highly active promoters belonging to immunoglobulin heavy chain genes causing upregulated and persistent expression of oncogenic *MYC* [15,92,93,94,95,96,97]. The chimeric ABL1-BCR protein has demonstrated the ability to transform benign cells into malignant cells, highlighting the necessity of understanding the mechanistic and evolutionary processes that result in the formation of such fusion genes [98].

Since their initial discovery and due to the advent of high-throughput sequencing a plethora of further gene fusions in malignancies have been identified and characterised (Table 1) [99,100,101,102]. A transcriptome RNA sequencing (RNA-seq) study—which allowed the detection of aberrant RNA transcripts across 675 commonly used cancer cell lines—uncovered 2200 gene fusion events, with a median of 3 fusions per cell line and 120 fusions that were found more than once [103,104]. Interestingly, the majority of these represented novel fusion transcript events that had not been observed previously. However, among the pairs of genes that constitute these 2200 fusions, 359 5${}^{\prime }$-partners and 238 3${}^{\prime }$-partners had previously been observed in other gene fusions with different partner genes, and 1822 fusions had one partner than could be observed in other fusions in RNA-seq data from TCGA database. These gene fusion events are known to occur with variable frequency among different cancer types, with some cancers almost guaranteed to posses one or more gene fusions, whereas, by contrast, in other cancers they are exceedingly rare [15,104]. As of writing (15 July 2021) there are currently 32,721 unique gene fusions involving 14,019 genes represented in the Mitelman Database of Chromosome Aberrations and Gene Fusions in Cancer (MDCAGFC), which is regularly and manually curated [102]. Given the large proportion of human genes represented in this database it is likely that many of these unique fusion genes are formed from passenger structural variants and are unlikely to be oncogenic in nature, having simply arisen as a result of the large amount of chromosomal rearrangement facilitated by increased genomic instability during tumourigenesis.

**Figure 2 genes-12-01376-f002:**
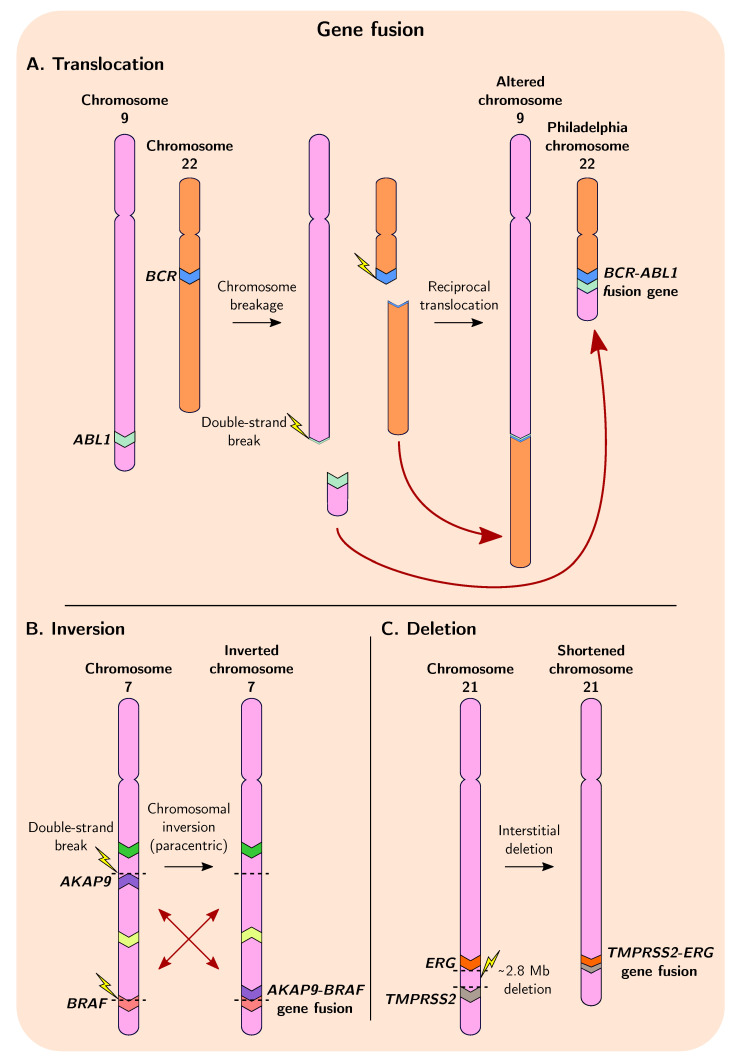
Overview of gene fusion in cancer with examples of well-known fusion events. (**A**) Reciprocal translocation between different chromosomes can result in gene fusion, such as during the well-characterised fusion of *BCR* and *ABL1* in CML. (**B**) Chromosomal inversion events can give rise to gene fusions, including the *AKAP9-BRAF* fusion found in thyroid carcinomas [105]. Inversion events can be pericentric (spanning the centromere) or paracentric (excluding the centromere) (**C**) Deletion of chromosomal segments between two genes can result in their fusion, which constitutes a large proportion of the fusion events forming *TMPRSS2-ERG* in prostate cancers.

**Figure 3 genes-12-01376-f003:**
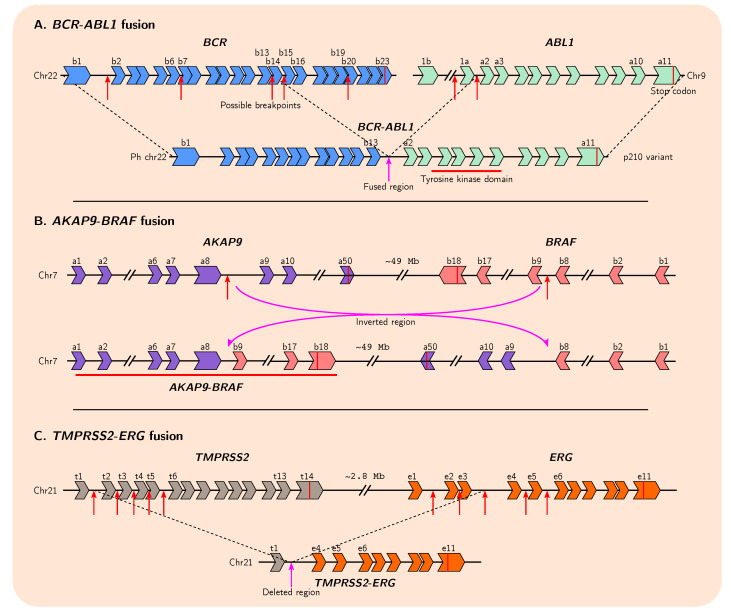
Snapshots of exon arrangements in well-known gene fusions in cancer. (**A**) Reciprocal translocation between different breakpoints within introns of *BCR* and *ABL1* can give rise to different fusion gene variants. (**B**) Chromosomal inversion between specific introns of *AKAP9* and *BRAF* results in fusion gene formation. (**C**) Breakage at multiple possible loci in *TMPRSS2* and *ERG* with subsequent interstitial deletion results in fusion of the two genes. Many fusion products are possible. Illustrations are simplified and are not to scale. Red arrows indicate breakpoints that have been previously experimentally identified.

Interestingly, two distinct genes can give rise to several possible fusion gene variant isoforms, as several breakpoint hotspots within intronic DNA have been identified for many fusion pairs (Figure 3A,C). This suggests that only certain regions of each gene are important for tumourigenesis; indeed, for some genes in many fusion pairs it appears only the initial promoter elements and first exon are required. In this manner, oncogenes which are ordinarily tightly controlled can gain more permissive expression by co-opting the promoters of other genes. This also may explain why fusion genes arise so frequently in cancer genomes, as the location of the breakpoints leading to their formation do not need to be so precise.

### 4.1. Tools for Detection of Fusion Genes

Several bioinformatic algorithms, including deFuse, FusionCatcher, PRADA, FusionHunter, SOAPfuse, JAFFA, STAR-Fusion, and Arriba, have been developed to detect fusion genes from RNA-seq and high-throughput sequencing (HTS) data to enable further cancer research and precision oncology therapy, and to filter out those fusions deemed to be non-functional [133,134,135,136,137,138,139,140,141]. The efficiency and accuracy of these tools vary, and many of them have been assessed and benchmarked to determine the best methods for accurate detection of fusion transcripts in cancer. From analyses performed on both simulated and cancer-cell-line-derived RNA-seq data, STAR-Fusion, and Arriba emerged as the most highly ranked algorithms with respect to prediction accuracy, while maintaining relatively fast execution times [141,142]. DeFuse, the most widely cited and one of the oldest programs, developed in 2011, performed comparatively poorly. However, different tools may situationally provide different advantages, depending on read length, and quantity and quality of the read data, so care should be taken when deciding on which software to utilise [139,143,144]. These tools have massively contributed to the plethora of gene fusions that have been identified in cancers thus far, represented in the MDCAGFC.

### 4.2. Role of Fusion Genes in Cancer Evolution

In some cancers fusion genes are thought to represent a crucial step in the initiation of tumourigenesis and contribute significantly to tumour burden and morbidity [145]. In hormone receptor-positive (HR${}^{+}$) breast cancer, tumours positive for rearrangement-mediated expressed gene fusions resulted in overall lower survival for patients relative to those with tumours negative for gene fusions [146,147,148]. Furthermore, certain oncogenes appear to require specific gene fusion events to become oncogenic, and most of these events show little variability in the type of structural variant that is capable of generating them [8]. In pilocytic astrocytomas, the *KIAA1549* and *BRAF* genes are frequently found fused as a result of tandem duplication, and in papillary thyroid cancers and thyroid carcinomas, inversion events are most often the cause of *RET-CCDC6* and *APAK9-BRAF* gene fusions, respectively (Figure 2B and Figure 3B) [105,128]. Additionally, specific chromosomal translocations common in follicular lymphomas juxtapose *BCL2* with the *IGH* immunoglobulin locus, causing overexpression of the anti-apoptotic *BCL2* and contributing to tumourigenesis [120,121]. Furthermore, fusion genes can also produce circular RNAs (circRNAs)—including *EML4*-*ALK1* and *EWSR1*-*FLI1*—that promote cellular transformation and tumour growth [125]. The overall relevance of circRNAs to tumourigenesis remains unclear, though it is possible these aberrant molecules are more oncogenically potent than their linear counterparts since circRNAs are generally thought to be more stable with longer half-lives, due to their resistance to exonuclease degradation [149,150,151].

Cancers in which fusion genes are commonly found, at least at levels detectable with current technologies, include prostate cancer and other cancers of the male genital organs, breast cancers, bone cancers, cancers of soft tissues and of the endocrine system, epithelial carcinomas, and some haematological malignancies [15,103,152,153,154]. Prostate cancer in particular, one of the most prevalent cancers in men worldwide, frequently exhibits several distinct fusion genes. For instance, fusions of *ETS* transcription factor gene *ERG* with promoter elements of *TMPRSS2* are observed in roughly 50% of prostate cancers (Figure 2C and Figure 3C), as well as *MTOR*-*TP53BP1*, *SLC45A2*-*AMACR*, and *MAN2A1*-*FER* fusions found at lower frequencies [89,129,131,155]. Although several studies suggest that presence of these fusion genes in prostate cancer, *TMPRSS2*-*ERG* in particular, is associated with more aggressive malignancy and worse clinical prognoses, not all studies agree [15,152,156,157,158,159,160,161]. However, these studies generally agree that there is a consistent association between presence of *TMPRSS2*-*ERG* and prostate cancer recurrence [131,152,157] In breast cancer, many gene fusion events have also been identified, with a recent genomic landscape study of adjacent gene rearrangements uncovering 99 recurrent gene fusions [162]. The most notable fusions involve microtubule-associated serine-threonine (*MAST*) kinase genes, which occur in a subset of around 3–5% of cases, and fusions involving *NOTCH* gene family members which occur in some HR${}^{-}$ breast cancers, with no overlap between cases positive for either fusion gene [103]. Overexpression of both *MAST1* and *MAST2* kinase-containing fusions in normal epithelial cells confers a proliferative advantage, and the presence of *NOTCH*-containing fusions in breast cancer cell lines leads to reduced cell-matrix adhesion, substantial morphological changes, and a unique vulnerability to *NOTCH* signaling inhibition, suggesting these fusion events, present in at least 5–7% of breast cancer cases overall, are most likely oncogenic in nature and contribute to tumourigenesis. Furthermore, it is thought that currently known gene fusions are responsible for at least 17–20% of morbidity in human cancer, in large part due to their high prevalence in prostate cancer [15].

Specific fusion genes can be observed recurrently both within and between distinct cancer types across different patients [89,115,163]. For instance, the *MAN2A1*-*FER* gene fusion, which activates *FER* tyrosine kinase—a well characterised oncogene that causes increased cellular proliferation via enhanced epidermal growth factor receptor activation—has been found in at least 6 distinct malignancies and is capable of inducing hepatocellular carcinomas by transforming normal liver cells [122,164]. This fusion gene exhibited 4-fold higher expression of *FER* kinase activity relative to wild-type expression. Furthermore, several fusion genes, such as *CCNH*-*C5orf30* and *TRMT11*-*GRIK2*, have been observed at variable frequencies across cancer cell lines and primary samples from 7 different malignancies including ovarian adenocarcinoma, oesophageal adenocarcinoma, hepatocellular carcinoma, non-small cell lung cancer, glioblastoma multiforme, breast cancer, and colon cancer [89]. Notably, the presence of *TRMT11*-*GRIK2* in ductal type breast cancers and in liver cancers was associated with more favourable outcomes, in contrast to prostate cancers positive for this fusion gene where patients exhibit worse clinical outcomes [131], suggesting fusion genes may have different effects depending on the cancer. However, the observation that the many of the same genes are commonly found across multiple fusion pairs in cancers (e.g., *BRAF*, *FGFR*, *ABL1*, *NTRK1/3*) suggests that selection favours fusions that contain these genes when they arise due to the competitive advantages they confer on their host cells [165,166,167,168,169]. In fact, many recurrent oncogeneic fusions that have been shown to drive cancer evolution—such as *BCR*-*ABL1* in CML—involve constitutive kinase expression or activation, which enhances downstream signalling pathways and elevates the rate of cell division [170,171,172]. Other common mechanisms of oncogenesis involve fusions which promote aberrant transcription, such as *TMPRSS2*-*ERG* seen across many prostate cancers [129,155].

### 4.3. Therapeutic Relevance of Fusion Genes

Proteins and transcripts from fusion genes can be used as diagnostic markers and can be therapeutically targeted in cancer treatment [172]. Since fusion genes are usually highly specific to cancer cells and not normally found in normal cells, targeting them may minimise off-target, debilitating side effects, which commonly occur during cancer treatment and decrease quality of life for many patients [173]. Indeed, several effective therapies exist that target specific oncogenic fusion genes (Table 2), including imatinib (Gleevec) for *BCR*-*ABL* fusions in haematological malignancies, and NVP-TAE684 for fusions involving *ALK* in anaplastic large-cell lymphomas and non-small-cell lung cancers [126,174,175,176,177]. Each of these drugs are receptor tyrosine kinase (RTK) inhibitors, suggesting that RTKs that form fusion genes may be selected for in certain cancers due to their strong oncogenic properties, given that targeting them appears particularly effective. Additionally, larotrectinib and entrectinib are recently approved drugs capable of targeting gene fusions containing neurotrophic *RTK* (*NRTK*) genes, which are commonly found in certain rare cancer types and in small subsets of many more common cancer types (Table 2) [168,178]. However, while these drugs can be quite effective at initially reducing tumour burden and morbidity they are, by their nature, environmental forces that exert potent selective pressure and can facilitate evolution and expansion of therapy-resistant cancer clones, thus simply delaying tumour progression rather than curing the cancer [9,172,179,180]. Recent theoretical studies have investigated the idea that tumour containment, not elimination, may be more beneficial to patient survival and quality of life [181]. These studies suggest that, by using a lower drug dosage regimen to reduce tumour burden instead of the maximum tolerated dose to try eliminate it completely, drug resistant clones are less likely to arise with a containment strategy and thus tumour growth can be restricted to a manageable level, potentially leading to more favourable outcomes. However, clinical trials using this strategy have not been attempted and so its effectiveness is, as of yet, unknown. Clearly, while gene fusions remain attractive therapeutic targets, further research on the evolution of cancer resistance to therapy is necessary.

Overall fusion genes represent powerful drivers of tumourigenesis and cancer evolution. Gene fusion events can bring together unrelated genes to form novel functions or enhance the function of one of the fusion partners, with much of the novelty often originating from altering or deregulating gene expression to enhance cellular proliferation and survival. The evolution of gene expression regulation is thought to represent an important molecular evolutionary mechanism by which phenotypic differences arise between species over time [187,188].

## 5. Conclusions

It is clear that gene duplication and gene fusion represent significant drivers of cancer evolution, the extent to which is becoming increasingly apparent with the ever expanding plethora of genomic data that has been generated in recent years. Genomic events and alterations that facilitate extensive gene duplication and fusion—including macroevolutionary events like whole-genome duplication and chromothripsis—are frequently beneficial to tumourigenesis and are positively selected for time and time again, often arising early in cancer formation. Given their significant and wide-ranging effects, elucidating the mechanisms behind these processes is crucial and will help us gain a greater understanding of cancer evolution on a somatic cellular level, allowing for the development of more effective therapeutic approaches to treating human malignancies.

## Figures and Tables

**Figure 1 genes-12-01376-f001:**
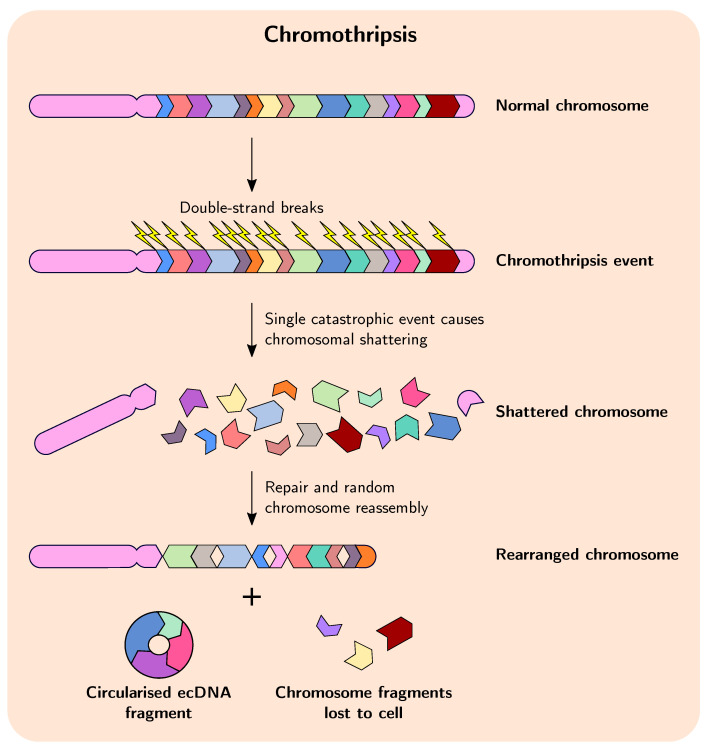
Overview of the process of chromothripsis in cancer. A single massive shattering event breaks up one or more chromosomes into multiple segments, which are then repaired and reassembled randomly to form new, rearranged chromosomes. Some segments can form circularised extracellular DNA fragments, and some can be lost to the cell.

**Table 1 genes-12-01376-t001:** List of common fusion genes identified in cancer.

Cancer Type	Fusion Gene(s)	References
Acute lymphoblastic leukaemia	*BCR-ABL1*${}^{*}$, *ETV6-RUNX1*, *TCF3-PBX1*	[106,107,108]
Acute megakaryoblastic leukaemia	*RBM15–MKL1* ${}^{*}$	[109]
Acute myeloid leukaemia	*RUNX1-RUNX1T1*(*AML1-MTG8*), *PML-RARA*, *CBFB-MYH11*, *BCR-ABL1*${}^{*}$, *RBM15–MKL1*${}^{*}$	[15,110,111,112,113]
Anaplastic large T-cell lymphoma	*NPM1–ALK*	[114]
Breast carcinoma	*TRMT11-GRIK2*${}^{*}$, *CCNH-C5orf30*${}^{*}$, *ETV6–NTRK3*${}^{*}$, *ODZ4–NRG1*, *TBL1XR1–RGS17*, *MYB-NFIB*${}^{*}$, *MAST*-fusions, *NOTCH*-fusions	[15,89,103,115]
Burkitt lymphoma	*IGH–MYC*, *IGK–MYC*, *IGL–MYC*	[92,116,117]
Chronic myeloid leukaemia	*BCR-ABL1* ${}^{*}$	[94]
Colorectal carcinoma	*TRMT11-GRIK2*${}^{*}$, *CCNH-C5orf30*${}^{*}$, *RSPO2-EIF3E*, *RSPO2-PTPRK*	[89,118]
Ewing’s sarcoma	*EWSR1-FLI1* ${}^{*}$	[119]
Fibrosarcoma	*ETV6–NTRK3* ${}^{*}$	[15]
Follicular lymphoma	*BCL2-IGH* ${}^{*}$	[120,121]
Glioblastoma multiforme	*TRMT11-GRIK2*${}^{*}$, *CCNH-C5orf30*${}^{*}$, *MAN2A1-FER*${}^{*}$, *FGFR3-TACC3*, *FIG-ROS1*${}^{*}$	[89,122,123,124]
Hepatocellular carcinoma	*TRMT11-GRIK2*${}^{*}$, *CCNH-C5orf30*${}^{*}$, *MAN2A1-FER*${}^{*}$	[89,122]
Lung cancer	*TRMT11-GRIK2*${}^{*}$, *CCNH-C5orf30*${}^{*}$, *EML4-ALK1*, *MAN2A1-FER*${}^{*}$, *FIG-ROS1*${}^{*}$	[89,122,124,125,126]
Oesophageal adenocarcinoma	*TRMT11-GRIK2*${}^{*}$, *CCNH-C5orf30*${}^{*}$, *MAN2A1-FER*${}^{*}$	[89,122]
Ovarian adenocarcinoma	*TRMT11-GRIK2*${}^{*}$, *CCNH-C5orf30*${}^{*}$, *MAN2A1-FER*${}^{*}$, *ESRRA-C11orf20*, *FIG-ROS1*${}^{*}$	[89,122,124,127]
Pilocytic astrocytoma	*BRAF-KIAA1549* ${}^{*}$	[128]
Prostate carcinoma	*TMPRSS2-ERG*, *TMPRSS2-ETV1*, *TMPRSS2-ETV4*, *MAN2A1-FER*${}^{*}$, *TRMT11-GRIK2*${}^{*}$, *SLC45A2-AMACR*, *TMEM135-CCDC67*, *MTOR-TP53BP1*, *CCNH-C5orf30*${}^{*}$, *RPS10–HPR*	[15,129,130,131]
Thyroid carcinoma	*APAK9-BRAF*${}^{*}$, *RET–CCDC6*, *PAX8–PPARG*, *TFG–NTRK1*, *TPM3–NTRK1*	[15,105,132]

* Recurring fusions across different cancer types.

**Table 2 genes-12-01376-t002:** List of therapeutically-targeted fusion genes.

Fusion Gene	Cancer Type(s)	Therapy/Drug	References
*BCR-ABL1*	Acute lymphoblastic leukaemia, acute myeloid leukaemia, chronic myeloid leukaemia	imatinib, axitinib, dasatinib, nilotinib, arsenic trioxide, ponatibib	[108,176,182,183,184,185]
*ALK*-fusions	Anaplastic large T-cell lymphoma	NVP-TAE684, crizotinib	[168,177,186]
*NRTK*-fusions	Secretory breast carcinoma, mammary analogue secretory carcinoma, congenital mesoblastic nephroma, infantile fibrosarcoma, thyroid cancer, melanoma, breast cancer	larotrectinib, entrectinib, LOXO-195, TPX-0005	[168,178]
*ROS1*-fusions	Non-small cell lung cancer	entrectinib	[124,168,177,186]
*PML-RARA*	Acute promyelocytic leukemia	All-trans retinoic acid, arsenic trioxide	[108]

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
