# Peer review of "Gene Duplication and Gene Fusion Are Important Drivers of Tumourigenesis during Cancer Evolution"

_genes, 2021, doi:10.3390/genes12091376_

Round 1

Reviewer 1 Report

This is a very nice review focusing on the functions/mechanisms of gene duplication and gene fusion and their potential contribution to cancer evolution. Both the historical perspective and information coverage are good, and it was fun to read the manuscript.

That said, one important aspect of current cancer evolution, the recently identified two-phased model of cancer evolution, is completely missing in this manuscript, giving me the impression that this paper could have been written years ago without the more recent appreciation of punctuated cancer evolution, where chromosomal changes are dominant. As pointed out by the authors, there are different types of genomic changes in most cancers, including karyotype changes, fusion genes, copy number variations, gene mutations, and epigenetic alterations. A newly emergent concept is karyotype coding: a concept where altered chromosomes can change the genetic network of the cell. Under this framework, chromothripsis, a type of genome chaos, can be understood both from a genome and gene perspective.

The authors can simply improve their presentation by inserting a paragraph to mention the progress in cancer evolutionary studies, as most cancer types do not follow the linear and stepwise evolution model. The following literature should be mentioned as well.     

PMID: 33930405.

PMID: 33189848.

PMID: 33649505.

PMID: 25665006.

PMID: 31737054.

Author Response

Thank you for your review and your positive response to our manuscript. We have revised the manuscript taking into account your comments, detailed below.

This is a very nice review focusing on the functions/mechanisms of gene duplication and gene fusion and their potential contribution to cancer evolution. Both the historical perspective and information coverage are good, and it was fun to read the manuscript.

That said, one important aspect of current cancer evolution, the recently identified two-phased model of cancer evolution, is completely missing in this manuscript, giving me the impression that this paper could have been written years ago without the more recent appreciation of punctuated cancer evolution, where chromosomal changes are dominant. As pointed out by the authors, there are different types of genomic changes in most cancers, including karyotype changes, fusion genes, copy number variations, gene mutations, and epigenetic alterations. A newly emergent concept is karyotype coding: a concept where altered chromosomes can change the genetic network of the cell. Under this framework, chromothripsis, a type of genome chaos, can be understood both from a genome and gene perspective.

The authors can simply improve their presentation by inserting a paragraph to mention the progress in cancer evolutionary studies, as most cancer types do not follow the linear and stepwise evolution model. The following literature should be mentioned as well.     

While we briefly mentioned the concept of punctuated evolution as an important step during cancer evolution in our original manuscript, we appreciate the need for additional details outlining the two-phase concept.  We have included additional details in the Chromothripsis section as such. Please see revised manuscript for the highlighted changes.

PMID: 33930405.

PMID: 33189848.

PMID: 33649505.

PMID: 25665006.

PMID: 31737054.

We have included the recommended literature.

Reviewer 2 Report

Comments of Glenfield and Innan:

In this review article, the authors summarized the role of both gene duplication and gene fusions in tumorigenesis and drug resistance. Combining both of these topics into one review provides a rather unique insight into how genomic instability can manifest in malignancy and cancer treatment. The information is presented in an overall thoughtful and nuanced way that is clear to understand and scientifically sound. The review starts with a discussion of the mechanisms related to gene duplication. In this section, the focus on chromothripsis is intriguing, especially the description of how the field has evolved to understand that genome instability and cancer development are two way-streets, not a strict casual relationship as has been traditionally understood. This is further expanded upon in the section 3, where the frequencies of structural variations (such as chromothripsis) across different cancer lines are discussed.

Regarding gene fusions, the authors have presented a brief discussion of how gene fusions lead to cancer as well as some of the technology used for the detection of fusion genes. This section may benefit from perhaps a table highlighting the most common gene-fusions across different cancer types. A table lists proven gene fusions that are drivers of tumorigenesis should also be helpful. They also describe how gene fusions are being used as therapeutic targets.

 Overall this review would allow a researcher not familiar in this area of study to develop a solid understanding of the current field of genomic alterations and cancer origins. In addition, the figures provided in the article are presented clearly and accurately depict the mechanistic processes mentioned by the authors. I recommend its acceptance with the following changes:

  1. In the title of the article, change “significant” to “important”.

  1. Line 1: chromosome rearrangement is not a hallmark of cancers (Hanahan and Weinberg, 2000, 2011)

  1. In the introduction of section 2, it would be worthwhile to emphasize how certain secondary chromosomal rearrangements can make it difficult to distinguish when gene amplifications are the actual drivers of cancer growth. Haber and Debatisse wrote a good summary about this issue in their 2006 minireview “Gene Amplification: Yeast Takes a Turn”.  This could also be added to section 3 (lines 216-220) when the authors discuss disentangling gene amplifications which drive cancer evolution. 

  1. In section 3, the last paragraph (lines 226 to 251), it would be worth referencing the review “Context is everything: aneuploidy in cancer”, as it expands on many of the subjects the authors have touched upon. Additionally, one thing that is mentioned in this review is the role of the immune system and the potential development of an immune tolerance to aneuploidy and could explain some of the observed differences across different tissues (including the ones highlighted in this section of the review). 

  1. In lines 347, the specific citation(s) that serve as evidence comparing the morbidities of non-fusion versus fusion originated cancers should be provided. 

  1. In line 407, I would recommend that a more recent paper or review article should be mentioned that captures the recent developments of acquired resistance to TKIs since 2017 given the rapidly evolving landscape, for example “Acquired Resistance to targeted therapies in NSCLC: updates and evolving sites” by Meador and Hata, 2020.

Author Response

Thank you for your review and your positive response to our manuscript. We have revised the manuscript taking into account your comments, detailed below.

In this review article, the authors summarized the role of both gene duplication and gene fusions in tumorigenesis and drug resistance. Combining both of these topics into one review provides a rather unique insight into how genomic instability can manifest in malignancy and cancer treatment. The information is presented in an overall thoughtful and nuanced way that is clear to understand and scientifically sound. The review starts with a discussion of the mechanisms related to gene duplication. In this section, the focus on chromothripsis is intriguing, especially the description of how the field has evolved to understand that genome instability and cancer development are two way-streets, not a strict casual relationship as has been traditionally understood. This is further expanded upon in the section 3, where the frequencies of structural variations (such as chromothripsis) across different cancer lines are discussed.

Regarding gene fusions, the authors have presented a brief discussion of how gene fusions lead to cancer as well as some of the technology used for the detection of fusion genes. This section may benefit from perhaps a table highlighting the most common gene-fusions across different cancer types. A table lists proven gene fusions that are drivers of tumorigenesis should also be helpful. They also describe how gene fusions are being used as therapeutic targets.

We have included two additional tables in the section on gene fusions, which we hope will be helpful. The first table outlines some of the common fusion genes found across a diverse array of cancers, while the second table outlines fusion genes for which therapies have been developed.

Overall this review would allow a researcher not familiar in this area of study to develop a solid understanding of the current field of genomic alterations and cancer origins. In addition, the figures provided in the article are presented clearly and accurately depict the mechanistic processes mentioned by the authors. I recommend its acceptance with the following changes:

  1. In the title of the article, change “significant” to “important”.

         We have made the requested change.

  1. Line 1: chromosome rearrangement is not a hallmark of cancers (Hanahan and Weinberg, 2000, 2011)

        We have revised the wording accordingly.

  1. In the introduction of section 2, it would be worthwhile to emphasize how certain secondary chromosomal rearrangements can make it difficult to distinguish when gene amplifications are the actual drivers of cancer growth. Haber and Debatisse wrote a good summary about this issue in their 2006 minireview “Gene Amplification: Yeast Takes a Turn”.  This could also be added to section 3 (lines 216-220) when the authors discuss disentangling gene amplifications which drive cancer evolution. 

         We have included the relevant discussion in section 3, as requested.

  1. In section 3, the last paragraph (lines 226 to 251), it would be worth referencing the review “Context is everything: aneuploidy in cancer”, as it expands on many of the subjects the authors have touched upon. Additionally, one thing that is mentioned in this review is the role of the immune system and the potential development of an immune tolerance to aneuploidy and could explain some of the observed differences across different tissues (including the ones highlighted in this section of the review). 

         We have included the recommended literature and provided a brief             discussion on the role of the immune system in governing the                          presence of aneuploidy in different cell types.

  1. In lines 347, the specific citation(s) that serve as evidence comparing the morbidities of non-fusion versus fusion originated cancers should be provided. 

         On further research we came across articles which disagree on this              point. We have revised the manuscript accordingly.

  1. In line 407, I would recommend that a more recent paper or review article should be mentioned that captures the recent developments of acquired resistance to TKIs since 2017 given the rapidly evolving landscape, for example “Acquired Resistance to targeted therapies in NSCLC: updates and evolving sites” by Meador and Hata, 2020.

         We have referenced some more recent literature in this section, as                requested.